# IMPROVING HYPERPARAMETER OPTIMIZATION BY PLANNING AHEAD

## ABSTRACT

Hyperparameter optimization (HPO) is generally treated as a bi-level optimization problem that involves fitting a (probabilistic) surrogate model to a set of observed hyperparameter responses, e.g. validation loss, and consequently maximizing an acquisition function using a surrogate model to identify good hyperparameter candidates for evaluation. The choice of a surrogate and/or acquisition function can be further improved via knowledge transfer across related tasks. In this paper, we propose a novel transfer learning approach, defined within the context of model-based reinforcement learning, where we represent the surrogate as an ensemble of probabilistic models that allows trajectory sampling. We further propose a new variant of model predictive control which employs a simple look-ahead strategy as a policy that optimizes a sequence of actions, representing hyperparameter candidates to expedite HPO. Our experiments on three meta-datasets comparing to state-of-the-art HPO algorithms including a model-free reinforcement learning approach show that the proposed method can outperform all baselines by exploiting a simple planning-based policy.

## 1 INTRODUCTION

Hyperparameter optimization (HPO) is a ubiquitous problem within the research community and an integral aspect of tuning machine learning algorithms to ensure generalization beyond the training data. HPO is often posed as a sequential decision-making process, however, it can be seen as a special use-case of model-based reinforcement learning (MbRL) (Sutton, 1991a; Henaff et al., 2017) developed under the guise of some idiosyncratic terms.

In MbRL the objective is to train a *transition model* to approximate an underlying transition function via interactions with an environment governed by some *policy*, e.g. random shooting (Nagabandi et al., 2018), to improve sample efficiency and learn from more useful interactions whereby the learned models are used as a simulator for sampling trajectories. An agent navigates the simulated environment to optimize a pre-defined *reward function*, while the transition model remains unchanged. Conventionally in HPO, a *surrogate model* is trained to estimate some black-box function, e.g. validation loss of a machine learning algorithm under investigation (Rasmussen, 2003; Snoek et al., 2015b; Springenberg et al., 2016). An *acquisition function* (Wilson et al., 2017b) interacts with the surrogate model to propose potential hyperparameters that optimize the black-box response, viewed as a *reward function*. Effectively, the surrogate model **is** the only unknown component for a transition model, that prevents HPO from being framed fully as MbRL problem.

In this paper, we present a novel formulation for HPO defined within the context of MbRL. Namely, we learn an ensemble of probabilistic neural network models (Lakshminarayanan et al., 2017) and show that using model predictive control (MPC) (Kamthe & Deisenroth, 2018) and a novel look-ahead variant to navigate the simulated black-box environment, we can outperform conventional Bayesian optimization techniques with heuristic acquisition functions in both transfer and non-transfer learning settings. Thus, we elaborate on the importance of *explicit* planning in HPO that has been largely overlooked by the community. We also formally define HPO as a Markov decision process (MDP) with a simple, yet novel, state representation as the *set* of previously evaluated hyperparameters and their corresponding responses.

We argue that with a clearly defined transition model, we can replace the acquisition function with a simple policy that maximizes the reward across the simulated trajectories, and achieve better results

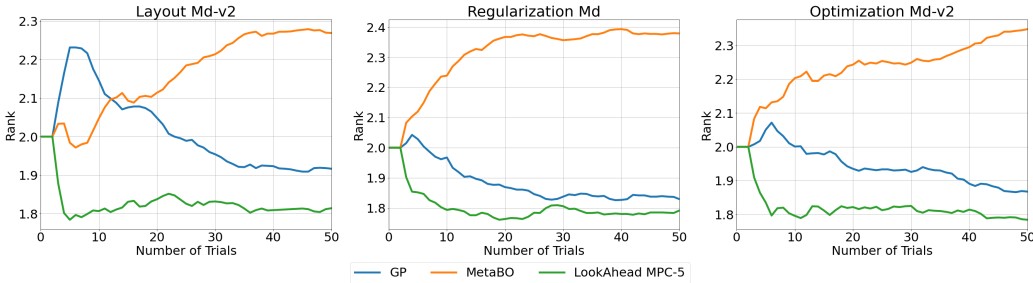

Figure 1: Motivating the effect of planning for HPO.

through proper planning, as shown in Figure 1. LookAhead MPC-5 is our look-ahead strategy with MPC that simulates 5 states ahead. We focus here on the motivation with more details in Section 6.

Our main contributions are summarized as i) a formal definition of HPO as an MDP which does not depend on any engineered heuristics, ii) a new transfer learning surrogate model represented by an ensemble of probabilistic neural networks, iii) a novel acquisition function that implements a look-ahead strategy paired with model predictive control, iv) clear motivation that highlights the impact of planning in HPO, which to the best of our knowledge, has never been addressed before.

## 2 RELATED WORK

Hyperparameter optimization has been extensively studied in the community within two main settings: single task (non-transfer learning) HPO and transfer learning HPO.

For single task HPO, solutions often involve estimating the hyperparameter response surface using a (probabilistic) surrogate, such as a Gaussian process (Rasmussen, 2003; Bergstra et al., 2011), random forests (Hutter et al., 2011), Bayesian neural networks (Springenberg et al., 2016), or some hybrid approach (Snoek et al., 2015a). Hyperparameter candidates are then selected via an acquisition function (Wilson et al., 2017a), e.g. expected improvement, that satisfies some well-motivated assumptions and utilizes the statistics from the posterior to score each hyperparameter.

Transfer learning solutions on the other hand leverage available data from experiments on related tasks to expedite HPO. Simple solutions involve initializing the surrogate using hyperparameters that perform well on datasets that have similar meta-features (Feurer et al., 2014; Jomaa et al., 2021a), i.e. dataset statistics. The response surface can also be modeled jointly in a multi-task setting (Bardenet et al., 2013; Yogatama & Mann, 2014; Perrone et al., 2018). For example, Salinas et al. (2020) transforms the hyperparameter response into a similar distribution and learns a shared Gaussian Copula. Another way to achieve transfer learning is through a weighted combination of the surrogates (Wistuba et al., 2016; Feurer et al., 2018). More recently, it has been shown that meta-learning an initialization of the surrogate by training to estimate the response across the training tasks, and subsequently fine-tuning to the target task improves generalization and leads to better performance (Wistuba & Grabocka, 2021; Jomaa et al., 2021b). Additionally one can learn a transferable acquisition function (Wistuba et al., 2018; Jomaa et al., 2019; Volpp et al., 2020). Jomaa et al. (2019) propose to learn a policy based on Deep Q-learning (Watkins & Dayan, 1992) where they assume a temporal dependency between the hyperparameters and propose an LSTM (Hochreiter & Schmidhuber, 1997) to model it. Volpp et al. (2020) propose a transferable acquisition function, MetaBO, as a policy that is meta-trained on related tasks. As another model-free RL approach, MetaBO is trained by interacting with the environment and observes an engineered state representation that is heavily influenced by underlying surrogates. MetaBO however is very sensitive to the number of trials (Wistuba & Grabocka, 2021). We discuss the proposed MDPs and their relationship to our proposed approach in Section 4.

Model-based reinforcement learning aims to learn a model of the environment by estimating the next state given a state-action pair. It is applied with great success in areas such as robotics and video games. Existing methods similarly involve learning a (probabilistic) model of the dynamics (Chua et al., 2018; Nagabandi et al., 2018; Ko et al., 2007). Moreover, several methods improve the policy

in a Dyna style approach (Sutton, 1991a), by leveraging the generated trajectories from the model to mitigate the distribution shift.

## 3 PRELIMINARIES

**Hyperparameter Optimization:** Let $\mathcal{D}$ denote the space of all datasets and $\Lambda \in \mathbb{R}^N$ the space of hyperparameters associated with an unknown black-box function $\ell^{(D)} : \Lambda \to \mathbb{R}$ where $\ell^{(D)}$ represents the response $\ell^{(D)}(\lambda)$, e.g. validation loss, of a certain model under investigation trained on a dataset $D \in \mathcal{D}$ with hyperparameters $\lambda \in \Lambda$. For example, we might be interested in a neural network model where $\Lambda := \mathbb{R}^+ \times \mathbb{R}_0^+ \times \mathbb{N}$ represents the learning rate, dropout rate, and the number of layers, respectively. The objective of HPO is hence to find the optimal hyperparameter configuration $\lambda^* := \arg\min_{\lambda \in \Lambda} \ell^{(D)}(\lambda)$ given a fixed budget of $T$ trials.

HPO is commonly treated as sequential decision-making process, where a surrogate model $\hat{\ell} : \Lambda \to \mathbb{R}$ is iteratively fit to the history $\mathcal{H}_t^{(D)} := \{\lambda_i, \ell^{(D)}(\lambda_i)\}_{i=1}^t$ of evaluated hyperparameters and an acquisition function $\hat{w} : (\Lambda \times \mathbb{R})^* \to \Lambda$ is used to select the next candidate which minimizes the expected hyperparameter response:

$$\arg\min_{\hat{w}} \mathbb{E}_{D \sim \rho_{\mathcal{D}}} \ell^{(D)}(\{\lambda_1, \dots, \lambda_t\}), \tag{1}$$

where $\ell^{(D)}(\{\lambda_1, \dots, \lambda_t\}) := \min_{i \in \{1, \dots, t\}} \ell^{(D)}(\lambda_i)$, $\lambda_i := \hat{w}(\{(\lambda_j, \ell^{(D)}(\lambda_j)\}_{j=1}^{i-1})$ and $\rho_{\mathcal{D}}$ is some distribution over the available datasets. Among the variety of acquisition functions, the *expected improvement* is widely adopted (Mockus, 1974). Although HPO algorithms can be applied on both continuous and discrete search spaces, here we focus on discrete spaces because they allow for faster training in transfer learning settings (Schilling et al., 2016; Jomaa et al., 2021b).

**Model-based reinforcement learning:** MbRL uses interactions with the environment to learn a parameterized approximation of the underlying transition function which subsequently can be used for planning. The learning task can be formulated as a discrete-time Markov decision process (MDP) defined by the tuple $\langle \mathcal{S}, \mathcal{A}, \tau, r \rangle$ with state space $\mathcal{S}$, action space $\mathcal{A}$, unknown transition function $\tau : \mathcal{S} \times \mathcal{A} \to \mathcal{S}$ and reward function $r : \mathcal{S} \times \mathcal{A} \to \mathbb{R}$. The general approach in RL is to learn the optimal policy $\pi$ maximizing the expected cumulative discounted reward given by $J(\pi) := \mathbb{E}\left[\sum_{t=1}^T \gamma^t r(s_t, a_t) \mid s_t = \tau(s_{t-1}, a_{t-1}), a \sim \pi(s_t)\right]$ where $s_t \in \mathcal{S}$ and $a_t \in \mathcal{A}$ with discount factor $\gamma \in [0, 1]$. We note that the explicit reward function or an *approximate* reward model is necessary for planning actions under some learned transition model. During inference, the optimal sequence of actions is commonly presented as the solution to the following optimization problem:

$$\arg\max_{a_1, \dots, a_T} \sum_{t=0}^{T-1} \hat{r}\left(\hat{\tau}(s_t, a_t),\ a_{t+1}\right), \tag{2}$$

which leverages the transition model $\hat{\tau}$ to predict the next state $s_{t+1}$ given a state $s_t$ and action $a_t$.

## 4 HYPERPARAMETER OPTIMIZATION VIA MODEL-BASED RL

In the following we formulate HPO as an MDP which in turn facilitates the use of a variety of RL approaches to solve the problem. As a sequential decision-making process, hyperparameters $\lambda \in \Lambda$ are iteratively selected and evaluated forming a simple MDP. Particularly, the state space is defined as $\mathcal{S} := (\Lambda \times \mathbb{R})^*$ with

$$s_t^{(D)} := \left((\lambda_1, \ell_1^{(D)}), \dots, (\lambda_t, \ell_t^{(D)})\right) \tag{3}$$

where $\ell_i^{(D)} := \ell^{(D)}(\lambda_i)$ and action space $\mathcal{A} := \Lambda$. The ground truth transition function $\tau : \mathcal{S} \times \Lambda \to \mathcal{S}$ simply appends the new observations to the previous ones,

$$\tau(s_t^{(D)}, \lambda) := \left((\lambda_1, \ell_1^{(D)}), \dots, (\lambda_t, \ell_t^{(D)}), (\lambda, \ell^{(D)}(\lambda))\right) \tag{4}$$

for dataset $D \in \mathcal{D}$ and a ground truth reward function, $r := \mathcal{S} \times \Lambda \to \mathbb{R}$, that returns the loss reduction of the new hyperparameters over the best loss so far,

$$r(s_t^{(D)}, \lambda) := \max\{0, \min_{i=1:t} \ell_i^{(D)} - \min\{\ell_1^{(D)}, \dots, \ell_t^{(D)}, \ell^{(D)}(\lambda)\}\} \tag{5}$$

To estimate both, the transition function and the reward function, as done in MbRL, the only missing piece is a model for the validation loss, the so called surrogate model. Given $\hat{\ell}^{(D)} : \Lambda \to \mathbb{R}$, the transition and reward models are just: $\hat{\tau} : \mathcal{S} \times \Lambda \to \mathcal{S}$ where

$$\hat{\tau}(s_t^{(D)}, \lambda) := \left( (\lambda_1, \ell_1^{(D)}), \ldots, (\lambda_t, \ell_t^{(D)}), (\lambda, \hat{\ell}^{(D)}(\lambda)) \right) \tag{6}$$

and $\hat{r} : \mathcal{S} \times \Lambda \to \mathbb{R}$ with

$$\hat{r}(s_t^{(D)}, \lambda) := \max\{0, \min_{i=1:t} \ell_i^{(D)} - \min\{\ell_1^{(D)}, \ldots, \ell_t^{(D)}, \hat{\ell}^{(D)}(\lambda)\}\}. \tag{7}$$

In this sense, HPO using a surrogate model can be seen as a special case of MbRL, and thus all approaches researched for MbRL can be directly applied to HPO. When learning a joint surrogate across several datasets in **D**, the superscript is dropped and $\hat{\ell}^{(D)} \to \hat{\ell}$.

To the best of our knowledge, so far there exist two prior publications which attempt to solve HPO using RL by defining distinct MDPs. Jomaa et al. (2019) originally define the state as the dynamic history of evaluated hyperparameters and their respective responses as $\mathcal{S} := (\Lambda \times \mathbb{R})^*$ and $s_t^{(D)}$ as defined earlier. However, they suggest that the order by which the hyperparameters are selected impacts the decision on which action to take next, which is why they model this temporal aspect via an LSTM in their policy, which is conditioned on some engineered meta-features. Their transition function generates new states by appending the action and the observed response to the previous state according to $\tau(s_t^{(D)}, \lambda) := \left( (\lambda_1, \ell_1^{(D)}), \ldots, (\lambda_t^{(D)}, \ell_t), (\lambda, \ell^{(D)}(\lambda)) \right)$. Volpp et al. (2020) represent the state space as $\mathcal{S} := (\mathbb{R}^M)^*$ with $s_t := ((\mu_t(\lambda_i), \sigma_t^2(\lambda_i), \lambda_i, \psi_i))_{i=1}^{|\Lambda|}$ where $\mu_t$ and $\sigma_t^2$ are the mean and variance of the posterior distribution of an underlying surrogate and $\psi_i$ are some engineered attributes. The transition function simply updates the parameters of the surrogate based on the new observations. Both approaches define the action space as the discrete grid of hyperparameters, $\mathcal{A} := \Lambda$, whereas the reward function is computed in terms of the regret.

In strong contrast to these methods that replace the standard acquisition function with a policy trained via model-free RL approaches, we adopt the powerful yet simple MDP proposed by Jomaa et al. (2019) to train a transition model that allows planning to improve HPO in the context of MbRL.

## 5 METAPETS ALGORITHM

In this section, we present our model-based reinforcement learning algorithm. Namely, we define the transition function, our novel look-ahead acquisition function, and the associated training procedure.

### 5.1 PROBABILISTIC TRANSITION FUNCTION

Model-based reinforcement learning is mainly concerned with learning a parameterized model of the environment which approximates the underlying dynamics, or transition function, $\tau : \mathcal{S} \times \mathcal{A} \to \mathcal{S}$, such that the next state can be estimated given the current state-action pair. Following the standard RL notation, we denote by actions $a$ hyperparameters $\lambda$, i.e. $\ell^{(D)}(a) = \ell^{(D)}(\lambda)$. Accordingly, the choice of the model class plays an important role. A common approach to model the environment is through Gaussian processes (GP) (Rasmussen & Kuss, 2003; Ko et al., 2007; Boedecker et al., 2014) or a mixture of GPs (Khansari-Zadeh & Billard, 2011) which provide uncertainty estimates for the predictions and, more importantly, for unexplored areas. However, such models suffer from the curse of dimensionality and thus are difficult to scale to high-dimensional domains. Deep neural networks on the other hand have shown great success in handling uncertainty estimation (Springenberg et al., 2016; Gal & Ghahramani, 2016; Lakshminarayanan et al., 2017), with success in predictive modeling of images (Watter et al., 2015) and short-horizon control tasks for high-dimensional data.

In our case, the state is represented as the history of selected hyperparameters and their responses, up to a given time, while the actions consist of the available hyperparameters to be evaluated. Hence there is only one missing piece of information that is required to establish the next state, and that is the response of the respective selected action (Figure 2). This missing element we acquire by training the transition model to directly estimate the response for a given state-action pair.

Specifically, we parameterize our transition function $\hat{\tau}_\theta(s_t^{(D)}, a_t) : \mathcal{S} \times \mathcal{A} \to \mathbb{R}_0^+ \times \mathbb{R}^+$ as a *probabilistic* neural network with parameters $\theta$, which takes as input the current state-action pair $(s_t^{(D)}, a_t)$ and outputs the defining parameters of a probability distribution function, e.g. a Gaussian distribution. The governing assumption for this well-established approach is that the instance is derived from a heteroscedastic Gaussian distribution, i.e. the variance is not fixed between instances. Accordingly, the network outputs the respective mean $\hat{\mu}_\theta \in \mathbb{R}_0$ and variance $\hat{\sigma}_\theta^2 \in \mathbb{R}^+$ and is trained by minimizing the negative log-likelihood:

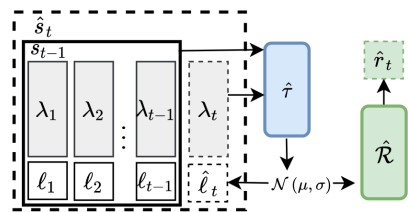

Figure 2: Architecture of the transition function

$$\mathbb{E}_{D \sim \rho_{\mathcal{D}}} \left[ -\frac{\log \hat{\sigma}_\theta(s^{(D)}, a)^2}{2} + \frac{(\ell^{(D)}(a) - \hat{\mu}_\theta(s^{(D)}, a))^2}{2\hat{\sigma}_\theta(s^{(D)}, a)^2} + \mathrm{ct} \right]. \tag{8}$$

Thus, we define the *predicted* hyperparameter response as $\hat{\ell}(a) \sim \mathcal{N}\left(\hat{\mu}_\theta(\cdot, a), \hat{\sigma}_\theta^2(\cdot, a)\right)$. We then estimate $\hat{s}_t^{(D)} = \hat{\tau}_\theta(s_t^{(D)}, a) = s_{t-1}^{(D)} \bigcup (a, \hat{\ell}(a))$. A simulated rollout is presented in Algorithm 1.

**The Case for Ensembles:** Probabilistic neural networks can capture and model the *aleatoric* uncertainty of the environment (e.g. observation noise) by parameterizing a suitable distribution. However, a single probabilistic network is not capable of capturing any *epistemic* uncertainties (e.g. lack of a sufficient amount of data to completely determine the real dynamics). In particular, the epistemic uncertainty vanishes in the limit of infinite data but can have a major impact in low data regimes like HPO. For that reason, it is of major importance to account for epistemic uncertainty.

A principled approach to deal with this issue is by bootstrapping models in an ensemble. An ensemble of deterministic models, such as standard neural networks, reduces bias and epistemic uncertainty. Therefore, to capture both, aleatoric as well as epistemic uncertainty, we employ a simple bootstrapped ensemble of probabilistic neural networks, which has been shown to outperform single probabilistic networks as well as other Bayesian approaches (Chua et al., 2018).

Given such an ensemble of probabilistic transition models denoted as $\{\hat{\tau}_{\theta_i}\}_{i=1}^{N_E}$, we can observe a mixture of $N_E$ distributions, i.e. $\{\mathcal{N}\left(\hat{\mu}_{\theta_i}, \hat{\sigma}_{\theta_i}^2\right)\}_{i=1}^{N_E}$. Following Lakshminarayanan et al. (2017) we aggregate the outputs as

$$\mu_*(\cdot, a) = \frac{1}{N_E} \sum_{i=1}^{N_E} \hat{\mu}_{\theta_i}(\cdot, a), \text{ and } \sigma_*^2(\cdot, a) = \frac{1}{N_E} \sum_{i=1}^{N_E} \left(\hat{\sigma}_{\theta_i}^2(\cdot, a) + \hat{\mu}_{\theta_i}^2(\cdot, a)\right) - \mu_*^2(\cdot, a) \tag{9}$$

leading to a *predicted* hyperparameter response in the form of $\hat{\ell}(a) \sim \mathcal{N}\left(\mu_*(\cdot, a), \sigma_*^2(\cdot, a)\right)$. The explicit advantage of the ensemble approach in particular is the simple but excellent estimation of uncertainty which constitutes one of the main performance drivers in HPO in recent years.

## 5.2 TRAINING THE DYNAMICS MODEL

The first step in our MbRL approach is to generate a dataset of transitions as training dataset for our dynamics model. We denote by $\mathcal{E} := \{\left(D_n, \{\lambda_i, \ell^{(D_n)}(\lambda_i)\}_{i=1}^{T_{D_n}}\right)\}_{n=1}^{N}$ a meta-dataset of primary datasets, hyperparameters and their responses sampled from some unknown distribution of datasets $\rho_D$ and some unknown distribution of hyperparameters $\rho_\Lambda$. We generate our training dataset $\mathcal{E}^{\mathrm{train}}$ from the meta-dataset by sampling a quadruple $(D_n, s_t^{(D_n)}, a_t, \ell^{(D_n)}(a_t))$ of datasets $D_n$, sets of hyperparameters and their responses as states $s_t^{(D_n)}$, unobserved hyperparameters as actions $a_t$, and the missing element from the next state $s_{t+1}^{(D_n)}$ as the response to $a_t$, i.e. $\ell^{(D_n)}(a_t)$. This is similar to the usually used approach of implementing a random policy to generate the trajectories (Nagabandi et al., 2018). Although it is possible to augment the $\mathcal{E}^{\mathrm{train}}$ with trajectories generated using MPC, we noticed that it is **i)** time-consuming and **ii)** does not lead to a significant improvement compared to using simple random trajectories. We conjecture that this is due to the nature of our transition function. Specifically, because the sequence of hyperparameters selected (by any policy) does not

immediately affect the output of the transition model, i.e. $\tau(s_{t-1}^{(D)}, a) = \tau(s_t^{(D)}, a) \, \forall a \in \mathcal{A}$ but rather the output is only affected by the selected action.

Since our state $s_t$ is defined as the collection of previously evaluated hyperparameters and their corresponding responses, we employ a *deep set* (Zaheer et al., 2017) formulation to encode $s_t$ into a fixed-size vector representation. In contrast to the common approach of conventional model-based RL algorithms which are concerned with predicting the state difference $\Delta \hat{s}_{t+1}$, we estimate the response $\ell^{(D)}(a)$ to a given action $a$ while considering the state as context information (Kim et al., 2019). Concretely, the transition model is defined as :

$$\hat{\mu}, \hat{\sigma}^2 = f\left(\left[a_t, \frac{1}{t}\sum_i^t g\left([\lambda_i, \ell_i]\right)\right]\right), \tag{10}$$

such that $\hat{\tau} := f \circ g$ where $g := \Lambda \times \mathbb{R} \to \mathbb{R}^{N_g}$ and $f : \Lambda \times \mathbb{R}^{N_g} \to \mathbb{R}_0^+ \times \mathbb{R}^+$ are feed-forward neural network models, and $[\,]$ represents standard concatenation. Given the variety of primary datasets, we use first-order meta-learning (Nichol et al., 2018) to optimize the parameters of our transition model. We summarize the training procedure in Algorithm 2.

### 5.3 PLANNING WITH THE LEARNED TRANSITION FUNCTION

Given the learned dynamics model, we subsequently require an approach that can effectively leverage the acquired information to predict the optimal sequence of hyperparameters to evaluate. For example, Dyna-style algorithms (Sutton, 1991b) use the simulated experience to mitigate the distribution shift (Luo et al., 2019; Kurutach et al., 2018; Clavera et al., 2018) by integrating trajectories that have been generated via the policy to further train the transition model.

#### 5.3.1 MODEL PREDICTIVE CONTROL

For simplicity, here we consider the well-known approach of *model-predictive control (MPC)* which has been used in many complex control scenarios (Bouffard et al., 2012; Lenz et al., 2015; Amos et al., 2018). Furthermore, it is easy to implement and does not require any gradient computation.

The general idea of MPC is to solve an optimization problem in a specific horizon on top of the learned transition model to produce a sequence of actions. In particular we employ a simple *random shooting (RS)* (Zhou & Yan, 2014) technique to solve the following optimization objective:

$$\arg\max_{\{a_{t'}\}_{t'=t}^T} \hat{r}(\hat{s}_{t'}^{(D)}, a_{t'}). \tag{11}$$

Notice that since the reward is measured in terms of the regret we do not need to sum over the rewards at each simulated state, as presented in Equation 2, but simply observe the reward at the final state in the rollout.

The RS policy generates $K$ random action sequences with a horizon of length $H$ from a uniform distribution and evaluates them via the learned transition function. Then usually only the first action of the best candidate sequence is executed and the procedure is repeated from the new state. Finally, we also fine-tune the transition model(s) given the newly evaluated hyperparameters on the test datasets to mitigate the distribution shift, which is common practice in all Bayesian-based optimization techniques.

#### 5.3.2 LOOKAHEAD MPC

Conventional acquisition functions are agnostic to the order by which the previous hyperparameters have been selected, i.e. $\hat{a}(H_t) = \hat{a}(\phi(H_t))$, where $\phi$ is some permutation function since they are only concerned with the immediate improvement and do not account for how the optimization process evolves. MPC on the other hand normally selects the first action from the trajectory that had the highest reward at the end of the rollout. This can be slightly misleading as it implies that to arrive at the best hyperparameter

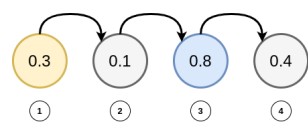

Figure 3: Simulated Trajectory rewards. MPC selects the first item. LookAhead MPC selects the third item.

configuration it is necessary to sequentially evaluate bad configu-
rations. However, this is not true since the selection of hyperpa-
rameter configurations is independent and all configurations can be
selected at all times. Thus, we propose to directly select the action
that provides the highest estimated reward, i.e. lowest regret, from all the actions observed across
all the simulated trajectories (Figure 3).

### 5.3.3 Trajectory Sampling

The learned transition model outputs a distribution over the next state $\hat{s}_{t+1}^{(D)}$. To leverage the uncer-
tainty associated with the output response, we create particles by randomly sampling $p$ variations as
$\hat{s}_{t+1}^{(D),p} \sim \hat{\tau}(s_t^{(D)}, a_t)$ and propagate through the trajectory of every particle. At each step, the reward
is then estimated as the average of the rewards across all *particles*:

$$\hat{r}(\hat{s}_t^{(D)}, a_t) = \frac{1}{P} \sum_{p=1}^{P} \hat{r}(\hat{s}_t^{(D),p}, a_t). \tag{12}$$

## 6 Experiments

The experiments are designed to address the following questions: 1)Does the learned transition
model generalize to new unseen tasks? 2) What is the importance of planning when doing HPO?

### 6.1 Meta-dataset

We evaluate our approach on three hyperparameter search spaces for feed-forward neural networks
(Jomaa et al., 2021b), that includes 120 UCI classification datasets (Asuncion & Newman, 2007).
We refer to the meta-datasets as **Layout Md-v2**, **Regularization Md**, and **Optimization Md-v2**
which include the Cartesian product of the individual hyperparameters with a total of 324, 288,
and 432 unique configurations, and 10, 7, and 12-dimensional hyperparameters, respectively. More
information is presented in Appendix B.

### 6.2 Training the transition model

The transition model is trained via first-order meta-learning. Each meta-dataset is divided into 5
splits with 80 training datasets, 16 for validation and 24 for testing. We used a task batch size of 8 and
64 mini-batches per task. The number of inner iterations was set to 5. We use the ADAM (Kingma &
Ba, 2015) optimizer with a learning rate of 0.001. The hyperparameters were tuned on the validation
set and trained for 10000 outer iterations with early stopping. We trained a total of 5 distinctly
initialized models for the ensemble using Tensorflow (Abadi et al., 2016).

### 6.3 Baseline Models

We compare against several single tasks and transfer learning baselines for HPO designed for black-
box function optimization that rely only on the hyperparameters and the corresponding response. As
such, some methods were not considered, e.g. Jomaa et al. (2021b); Falkner et al. (2018). Partic-
ularly, we compare against, Random sampling (Bergstra & Bengio, 2012), GP (Rasmussen, 2003),
SMAC (Hutter et al., 2011), BOHAMIANN (Springenberg et al., 2016), TST-R (Wistuba et al.,
2016), ABLR (Perrone et al., 2018), CTS (Salinas et al., 2020), and FSBO (Wistuba & Grabocka,
2021). A detailed overview can be found in Appendix C.

We denote by LookAhead MPC-H**X**, our approach, which employs LookAhead MPC by navigating
the simulated environment using random shooting up to the defined horizon **X** and then selects the
first action of the trajectory that achieves the highest reward. We sample 1000 trajectories for our
method. The models of the ensemble are iteratively fine-tuned to the new hyperparameters as all
baseline models. In Algorithm 3 we present the pseudo-code for our MetaPETS algorithm and
provide the source code in the Supplementary material.

| Methods \ Trials | Layout Md-v2 | | | Regularization Md | | | Optimization Md-v2 | | |
|---|---|---|---|---|---|---|---|---|---|
| | 15 | 33 | 50 | 15 | 33 | 50 | 15 | 33 | 50 |
| RS | $6.48 \pm 0.62$ | $6.81 \pm 0.38$ | $6.78 \pm 0.17$ | $7.11 \pm 0.49$ | $7.33 \pm 0.30$ | $7.23 \pm 0.21$ | $7.19 \pm 0.55$ | $6.93 \pm 0.44$ | $7.07 \pm 0.27$ |
| BOHAMIANN | $6.67 \pm 0.05$ | $6.48 \pm 0.26$ | $6.29 \pm 0.17$ | $6.51 \pm 0.22$ | $6.23 \pm 0.14$ | $5.86 \pm 0.16$ | $6.87 \pm 0.35$ | $6.55 \pm 0.26$ | $6.30 \pm 0.17$ |
| GP | $6.28 \pm 0.50$ | $5.73 \pm 0.19$ | $5.72 \pm 0.07$ | $5.48 \pm 0.19$ | $5.32 \pm 0.19$ | $5.45 \pm 0.16$ | $5.92 \pm 0.10$ | $5.76 \pm 0.11$ | $5.53 \pm 0.16$ |
| SMAC | $6.06 \pm 0.24$ | $6.19 \pm 0.32$ | $6.36 \pm 0.18$ | $6.22 \pm 0.30$ | $6.34 \pm 0.06$ | $6.43 \pm 0.19$ | $6.10 \pm 0.26$ | $6.24 \pm 0.28$ | $6.27 \pm 0.29$ |
| CTS | $5.66 \pm 0.25$ | $5.74 \pm 0.34$ | $5.89 \pm 0.22$ | $5.80 \pm 0.30$ | $5.80 \pm 0.15$ | $5.98 \pm 0.16$ | $5.73 \pm 0.26$ | $6.16 \pm 0.28$ | $6.18 \pm 0.27$ |
| ABLR | $7.11 \pm 0.50$ | $6.78 \pm 0.36$ | $6.55 \pm 0.27$ | $6.72 \pm 0.39$ | $6.36 \pm 0.43$ | $6.03 \pm 0.21$ | $5.80 \pm 0.21$ | $5.79 \pm 0.20$ | $5.72 \pm 0.25$ |
| TST-R | $5.54 \pm 0.23$ | $5.42 \pm 0.17$ | $5.49 \pm 0.13$ | $5.50 \pm 0.17$ | $5.48 \pm 0.09$ | $5.48 \pm 0.13$ | $5.75 \pm 0.20$ | $5.59 \pm 0.15$ | $5.57 \pm 0.13$ |
| MetaBO | $6.38 \pm 0.26$ | $6.85 \pm 0.12$ | $7.02 \pm 0.15$ | $7.02 \pm 0.13$ | $7.42 \pm 0.15$ | $7.60 \pm 0.09$ | $6.69 \pm 0.15$ | $7.00 \pm 0.08$ | $7.36 \pm 0.08$ |
| FSBO | $\underline{5.31} \pm 0.29$ | $5.35 \pm 0.27$ | $\underline{5.31} \pm 0.15$ | $5.50 \pm 0.14$ | $5.45 \pm 0.14$ | $5.56 \pm 0.17$ | $\underline{5.33} \pm 0.17$ | $5.39 \pm 0.18$ | $5.38 \pm 0.28$ |
| LookAhead MPC-3 | $\mathbf{5.18} \pm 0.16$ | $\mathbf{5.31} \pm 0.21$ | $\mathbf{5.26} \pm 0.08$ | $\underline{5.10} \pm 0.15$ | $\mathbf{5.13} \pm 0.12$ | $\mathbf{5.15} \pm 0.10$ | $5.35 \pm 0.15$ | $\underline{5.36} \pm 0.22$ | $\underline{5.36} \pm 0.24$ |
| LookAhead MPC-5 | $5.32 \pm 0.20$ | $\underline{5.35} \pm 0.08$ | $5.32 \pm 0.15$ | $\mathbf{5.03} \pm 0.14$ | $\underline{5.14} \pm 0.22$ | $\underline{5.23} \pm 0.11$ | $\mathbf{5.27} \pm 0.15$ | $\mathbf{5.23} \pm 0.27$ | $\mathbf{5.26} \pm 0.23$ |

Table 1: Average rank. We report the best results in **bold** and underline the second best.

| Methods \ Trials | Layout Md-v2 | | | Regularization Md | | | Optimization Md-v2 | | |
|---|---|---|---|---|---|---|---|---|---|
| | 15 | 33 | 50 | 15 | 33 | 50 | 15 | 33 | 50 |
| RS | $7.90 \pm 1.59$ | $5.59 \pm 0.98$ | $4.25 \pm 0.64$ | $8.54 \pm 0.84$ | $6.18 \pm 0.56$ | $4.51 \pm 0.30$ | $8.53 \pm 0.89$ | $5.40 \pm 0.39$ | $4.64 \pm 0.32$ |
| BOHAMIANN | $8.05 \pm 0.39$ | $4.78 \pm 0.17$ | $3.55 \pm 0.22$ | $7.59 \pm 0.39$ | $4.30 \pm 0.28$ | $2.64 \pm 0.40$ | $8.64 \pm 0.90$ | $5.19 \pm 0.48$ | $3.75 \pm 0.16$ |
| GP | $7.90 \pm 1.06$ | $4.25 \pm 0.26$ | $3.12 \pm 0.18$ | $6.05 \pm 0.27$ | $3.46 \pm 0.30$ | $2.49 \pm 0.19$ | $6.95 \pm 0.37$ | $4.51 \pm 0.17$ | $3.38 \pm 0.20$ |
| SMAC | $7.56 \pm 0.36$ | $4.68 \pm 0.47$ | $3.66 \pm 0.28$ | $7.31 \pm 0.77$ | $4.81 \pm 0.38$ | $3.73 \pm 0.44$ | $6.96 \pm 0.42$ | $4.61 \pm 0.33$ | $3.69 \pm 0.43$ |
| CTS | $6.39 \pm 0.54$ | $4.05 \pm 0.64$ | $3.32 \pm 0.38$ | $6.56 \pm 0.59$ | $3.82 \pm 0.24$ | $3.00 \pm 0.29$ | $6.67 \pm 0.26$ | $4.59 \pm 0.39$ | $3.58 \pm 0.22$ |
| ABLR | $9.01 \pm 0.88$ | $5.60 \pm 0.75$ | $4.26 \pm 0.44$ | $7.99 \pm 0.57$ | $4.98 \pm 0.52$ | $3.53 \pm 0.33$ | $7.30 \pm 0.33$ | $4.68 \pm 0.56$ | $3.64 \pm 0.55$ |
| TST-R | $6.70 \pm 0.36$ | $3.90 \pm 0.36$ | $2.91 \pm 0.33$ | $6.51 \pm 0.39$ | $3.88 \pm 0.29$ | $2.73 \pm 0.39$ | $6.78 \pm 0.21$ | $4.13 \pm 0.23$ | $3.21 \pm 0.10$ |
| MetaBO | $7.24 \pm 0.37$ | $4.72 \pm 0.32$ | $3.86 \pm 0.28$ | $7.71 \pm 0.11$ | $5.72 \pm 0.20$ | $5.01 \pm 0.21$ | $8.63 \pm 0.41$ | $5.83 \pm 0.24$ | $5.13 \pm 0.19$ |
| FSBO | $6.41 \pm 0.50$ | $\mathbf{3.60} \pm 0.31$ | $\mathbf{2.48} \pm 0.29$ | $5.99 \pm 0.57$ | $3.18 \pm 0.14$ | $2.32 \pm 0.26$ | $\mathbf{6.06} \pm 0.37$ | $\underline{3.90} \pm 0.30$ | $\underline{3.04} \pm 0.41$ |
| LookAhead MPC-3 | $\mathbf{6.15} \pm 0.49$ | $\underline{3.65} \pm 0.42$ | $\underline{2.60} \pm 0.32$ | $\underline{5.50} \pm 0.39$ | $\mathbf{2.95} \pm 0.24$ | $\mathbf{1.87} \pm 0.14$ | $6.18 \pm 0.73$ | $4.15 \pm 0.39$ | $3.16 \pm 0.30$ |
| LookAhead MPC-5 | $\underline{6.25} \pm 0.51$ | $3.90 \pm 0.21$ | $2.85 \pm 0.38$ | $\mathbf{5.34} \pm 0.09$ | $\underline{2.97} \pm 0.29$ | $\underline{1.91} \pm 0.18$ | $\underline{6.15} \pm 0.47$ | $\mathbf{3.77} \pm 0.67$ | $\mathbf{2.95} \pm 0.47$ |

Table 2: Average normalized regret. We report the best results in **bold** and underline second best.

## 6.4 EVALUATION METRICS

We report two performance metrics to evaluate the effectiveness of the different baselines: **normalized regret** that represents the distance between the response of the evaluated hyperparameters and the optimal performance for each dataset, and the **rank** that measures the relative performance of each method compared to the rest of the baselines at each trial. The rank is computed at the task level and is agnostic to the heterogeneous ranges of the target response surfaces, thus can better signify the difference in performance.

## 6.5 RESULTS AND DISCUSSION

We report in Tables 1 and 2 the average rank and normalized regret, respectively, over 5 runs for 50 trials with three different seeds per run for all methods. Each transfer learning approach had access to the same training datasets and all models were initialized via the same three seeds to fit the initial surrogate. Overall, our LookAhead method outperforms the baselines across all meta-datasets. We use the horizon of $H3$ and $H5$ in this section, and study the contribution of LookAhead and the importance of planning in Section 6.6.

LookAhead MPC-3 and LookAhead MPC-5 demonstrate clear gains in the average rank with consistent performance against the baselines. Looking at the normalized regret, FSBO outperforms our model in some cases, e.g. Layout Md-v2, however, the associated average rank is still lower. This can be attributed to the fact that different tasks have heterogeneous response distributions. When there is a clear contradiction between the rank and the normalized regret, this signifies that the margin of difference of the normalized regret on the *few* tasks where FSBO shows stronger performance, is higher than where it has lost.

Comparing LookAhead MPC-3 and LookAhead MPC-5, we notice that a shorter horizon is sufficient to explore smaller search spaces. Albeit, early exploration in the form of longer horizons can still turn out to be more favorable, e.g in Regularization Md.

It is also worth mentioning that MetaBO, the model-free RL approach, is the worst across all meta-datasets. This is attributed to the asymptotic degradation of the performance with the increasing number of trials, as pointed out in Wistuba & Grabocka (2021). Contrary to MetaBO, which relies on an underlying surrogate to generate the state which is bound by a fixed grid, our approach is

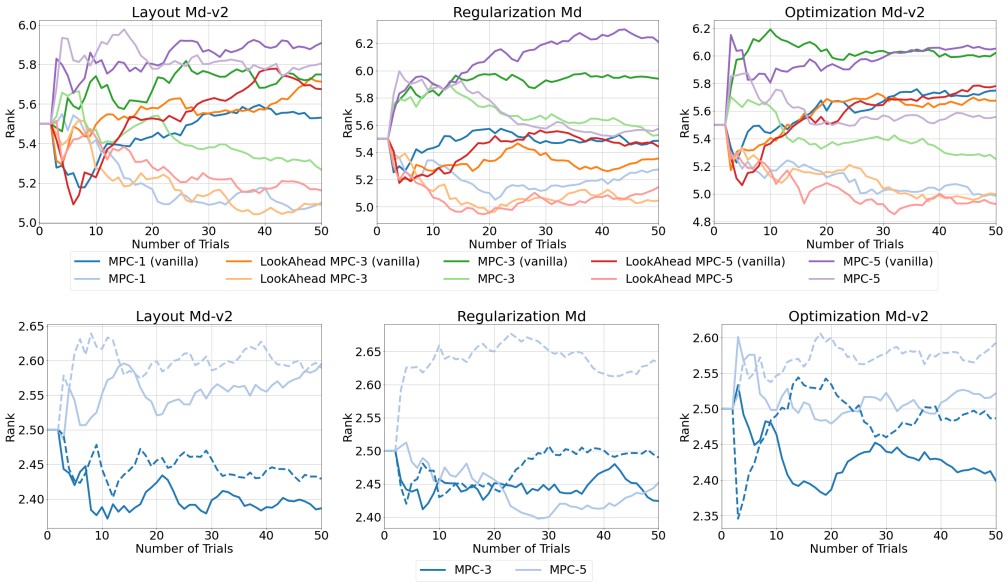

Figure 4: (top) Investigating the impact of planning and LookAhead across different horizons; (bottom) Increasing the number of sampled trajectories improves planning (solid lines→ 1000 trajectories vs. dashed lines → 100 trajectories).

agnostic to the size of the grid and improves with more trials, leading to an improved state representation.

## 6.6 ABLATION

We further investigate the effect of LookAhead on MPC with different horizons and trajectory samples to evaluate the impact of planning under such conditions. Moreover, we compare against a *vanilla* version, that does not involve fine-tuning the transition model to the observations of the target dataset. We summarize the performance in Figures 4 and share the following insights: i) Fine-tuning plays a crucial role in improving performance. Contrary to existing MbRL solutions that evaluate a policy's performance on the environment which the transition model has been trained to estimate, we evaluate our policy on new environments, i.e. new tasks, with varying response surfaces. We notice that with a few gradient update steps, (LookAhead) MPC-**X** outperforms the vanilla variant. ii) Using LookAhead MPC is better than standard MPC. This reinforces the notion that via proper planning the evaluation of bad hyperparameter configurations can be avoided. iii) Increasing the number of sampled trajectories leads to a better outcome. This is to be expected, considering that for random shooting, we sample a fixed number of trajectories to evaluate. At any given trial $t$, the number of possible trajectories for a horizon $H$ on a grid of size $N$ is $\binom{N-t}{H}$, so for a horizon $H = 1$, we default to maximization over the grid. In Appendix D, we further discuss the effect of various plausible dynamics by bootstrapping different sets of models as part of the ensemble.

## 7 CONCLUSION

In this paper, we present a novel solution for HPO within the context of MbRL. Specifically, we design a new surrogate as an ensemble of neural networks, which is initialized through meta-learning and thus can adapt quickly to new target datasets with few observations. We propose a novel acquisition function based on model predictive control that utilizes a simple lookahead strategy to select good hyperparameter candidates from simulated trajectories. To the best of our knowledge, we are the first to investigate the impact of planning on HPO and present an extensive ablation study that motivates research in that direction.

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

## A    ALGORITHM

---

**Algorithm 1** Simulated Rollout

---

1: **Require:** state $s_t^{(D)}$, rollout horizon $H$, transition model $\hat{\tau}_\theta$, actions $(a_h)_{h=0}^H$
2: $R \leftarrow \emptyset$
3: $\hat{s}_t^{(D)} \leftarrow s_t^{(D)}$
4: **for** $h = 0$ to $H$
5: $\quad \hat{\mu}_\theta, \hat{\sigma}_\theta^2 \leftarrow \hat{\tau}_\theta(\hat{s}_{t+h}^{(D)}, a_h)$
6: $\quad \hat{\ell}(a_h) \sim \mathcal{N}\left(\hat{\mu}_\theta, \hat{\sigma}_\theta^2\right)$
7: $\quad R \leftarrow R \bigcup(\hat{\ell}(a_h))$
8: $\quad \hat{s}_{t+h+1}^{(D)} \leftarrow \hat{s}_{t+h}^{(D)} \bigcup(a_h, \hat{\ell}(a_h))$
9: **return** $R$

---

Meta-learning has found resounding success in the research community as an initialization scheme, which allows for fast adaption to new domains. We want to emphasize that during meta-training the meta-test datasets are not observed and in that way remain strictly held out.

---

**Algorithm 2** Model-based Reinforcement Learning

---

1: **Require:** training meta-dataset $\mathcal{E}^{\text{train}}$, parameters $\theta$, learning rate $\eta$, inner update steps $v$, meta-batch size $n$, minimum history length $T_min$, maximum history length $T_max$
2: **while** not converged **do**
3: $\quad t \sim \text{Unif}\left([T_{\min}, T_{\max}]\right)$
4: $\quad D_1, \ldots, D_n \sim \text{Unif}([1, \ldots, N])$
5: $\quad$ **for** $i = 1$ to $n$ **do**
6: $\quad\quad \left(s_t^{(D_i)}, a_t, \ell^{(D_i)}(a_t)\right) \sim \text{Unif}\left(\mathcal{E}^{\text{train}} \mid D_i, t\right)$
7: $\quad\quad \theta_i \leftarrow \theta$
8: $\quad\quad$ **for** $j = 1$ to $v$ **do**
9: $\quad\quad\quad \theta_i \leftarrow \theta_i + \eta \nabla_\theta \left(p_\theta(s_t^{(D_i)} \mid s_t^{(D_i)}, a_t)\right)$
10: $\quad$ Update $\theta \leftarrow \theta + \eta \frac{1}{n} \sum_{i=1}^n (\theta_i - \theta)$
11: **return** $\theta$

---

## B    META-DATASET

A meta-dataset is a collection of hyperparameters, typically defined on a *discretized* grid (Schilling et al., 2016; Jomaa et al., 2019), associated with a model under investigation, that have been evaluated offline by training the model with the mentioned hyperparameters on numerous *primary* datasets, and reporting some observed evaluation metric, e.g. validation loss.

We evaluate our approach on three hyperparameter search spaces for feed-forward neural networks (Jomaa et al., 2021b), that includes 120 UCI classification datasets (Asuncion & Newman, 2007). Jomaa et al. (2021b) propose a pruning strategy that eliminates *redundant* configurations. Specifically, they drop certain hyperparameter combinations, e.g. non-□ layouts with 1 hidden layer, which we see here as unnecessary. Therefore we do not prune **Layout Md** and **Optimization Md** and refer to the meta-datasets as **Layout Md-v2**, **Regularization Md**, and **Optimization Md-v2** where each meta-dataset includes the Cartesian product of the individual hyperparameters with a total of 324, 288, and 432 unique configurations, respectively.

The aforementioned hyperparameters are encoded as follows:

## C    BASELINES

1. Random sampling (Bergstra & Bengio, 2012),

---

**Algorithm 3** MetaPETS Algorithm

---

1: **Require:** target dataset $D$; training meta-dataset $\mathcal{E}^{\text{train}}$, transition model $\hat{\tau}_\theta$, parameters $\theta$, learning rate $\eta$, inner update steps $v$, meta-batch size $n$, minimum history length $T_{min}$, maximum history length $T_{max}$, initial budget $b$, total budget $B$, planning horizon $H$, lookahead indicator $z$, vanilla indicator $w$
2: Initialize $\theta$ randomly
3: $\theta \leftarrow$ Algorithm $2\left(\mathcal{E}^{\text{train}}, \theta, \eta, v, n\right)$
4: $a_0, \ldots, a_b \sim \text{Unif}(\Lambda)$
5: $s_0^{(D)} \leftarrow \left((a_0, \ell_0^{(D)}), \ldots, (a_b, \ell_b^{(D)})\right)$
6: $a^* \leftarrow \arg\min_{\{a_i\}_{i=0}^b} \ell^{(D)}(a_i)$
7: **for** $i = 0$ to $B$ **do**
8:      Sample sequence of actions $(a_h^i)_{h=0}^H$ randomly
9:      Perform simulated rollout, $R \leftarrow$ Algorithm $1(s_{b+i}^{(D)}, H, \hat{\tau}_\theta, (a_h^i)_{h=0}^H)$
10:      $a_i \leftarrow \begin{cases} a_0^i & \text{if } z = 0 \text{ (selects first action from sequence)} \\ a_{\arg\min R}^i & \text{otherwise (selects best action from sequence)} \end{cases}$
11:      $a^* \leftarrow \begin{cases} a_i & \text{if } \ell^{(D)}(a_i) < \ell^{(D)}(a^*) \\ a^* & \text{otherwise} \end{cases}$
12:      **if** $w = 0$ **do**
13:          **while** *not converged* (Comment: Fine-tuning)
14:              **for** $t = 0$ to $b + i$
15:                  $\theta \leftarrow \theta + \eta \nabla_\theta \left(p_\theta(s_t \mid s_t, a_{t+1})\right)$
16:      $s_{b+i+1}^{(D)} \leftarrow s_{b+i}^{(D)} \bigcup (a_i, \ell(a_i))$
17: **return** $a^*$

---

Table 3: Hyperparameter search space for the meta-datasets.

| Hyperparameter | Layout Md-v2 | Regularization Md | Optimization Md-v2 |
|---|---|---|---|
| Activation | ReLU, SeLU | ReLU, SeLU, LeakyReLU | ReLU, SeLU, LeakyReLU |
| Neurons | $4, 8, 16, 32$ | $4, 8, 16, 32$ | $4, 8, 16$ |
| Layers | $1, 3, 5, 7$ | $1, 3, 5, 7$ | $3, 5, 7$ |
| Layout | $\square, \triangleleft, \triangleright, \diamond, \triangle$ | $\square$ | $\triangleleft, \triangleright, \diamond, \triangle$ |
| Dropout | $0, 0.5$ | $0, 0.2, 0.5$ | $0$ |
| Normalization | False | False, True | False |
| Optimizer | ADAM | ADAM | ADAM, RMSProp, GD |

2. GP (Rasmussen, 2003) is a hyperparameter tuning strategy that relies on a Gaussian process as a surrogate model with squared exponential kernels (Matern 5/2 kernel) with automatic relevance determination,

3. SMAC (Hutter et al., 2011) utilizes random forests instead of Gaussian processes to represent the surrogate model,

4. BOHAMIANN (Springenberg et al., 2016) is based on Bayesian neural networks that are trained via stochastic gradient Hamiltonian Monte Carlo,

5. TST-R (Wistuba et al., 2016) is an ensemble approach where the Gaussian process surrogate of the target task is weighted with surrogates of the training datasets based on the ranking similarity of the evaluated hyperparameters,

6. ABLR (Perrone et al., 2018) is a multi-task Bayesian linear regression approach that optimizes a shared feature extractor across the training datasets as an initialization strategy for the target task,

7. CTS (Salinas et al., 2020) trains a Gaussian Copula process (Wilson & Ghahramani, 2010) jointly over the training datasets mapped to a shared output distribution using quantile-transformations. Hyperparameter candidates are selected via Thompson Sampling,

Table 4: Hyperparameter Encoding

| Hyperparameter | Encoding |
|---|---|
| Activation | One-hot encoding |
| Neurons | Scalar |
| Layers | Scalar |
| Layout | One-hot encoding |
| Dropout | Scalar |
| Normalization | Scalar |
| Optimizer | One-hot encoding |

8. FSBO (Wistuba & Grabocka, 2021) uses deep Kernel Gaussian processes (Wilson et al., 2015) to estimate the response of the target dataset. The parameters are initialized via meta-learning the joint response surface over the training datasets.

## D    EVALUATING VARIOUS STATE PROPAGATION DYNAMICS

In the main experiment, we predict plausible state trajectories using a bootstrap of $B = 5$ models in the ensemble. We investigate here the impact of changing the models in the ensemble, i.e. bootstrapping subsets of these models, to understand the stability and robustness of probabilistic neural networks. We propose as a variant E$X$, where we re-sample at every trial uniformly at random a different subset of $B = X$ models. Effectively, we would be continually re-sampling from the approximate marginal posterior of plausible dynamics. We summarize the results in Figure 5.

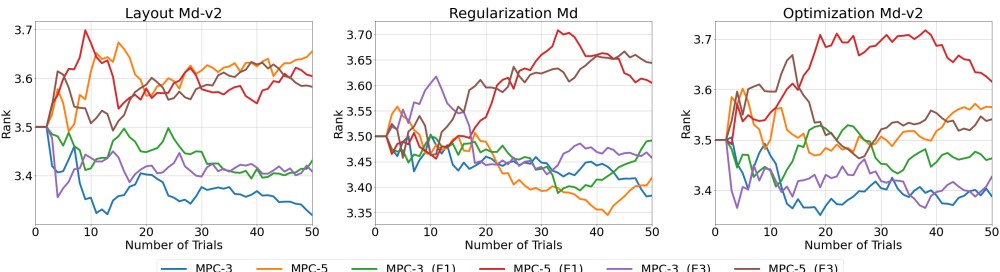

Figure 5: Investigating the performance of MPC under various dynamics.

We notice primarily that increasing that bootstrapping more models generally leads to better performance. In Layout Md-v2, MPC-3 and MPC-3 (E3) are better than MPC-3 (E1), especially at the early stages. We also notice that increasing the horizon leads to worse results. The same behavior is noticeable with Optimization Md-v2. Although MPC-5 in Regularization Md is better than the rest, discussed in Section 6.6, we notice that with the increasing number of trials, re-sampling a single probabilistic neural network starts to deteriorate.

## E    LIMITATIONS

The proposed MetaPETS algorithm is designed as a transfer learning solution for HPO. The transition model is trained on a set of correlated source tasks to approximate the joint response surface. One of the limitations of this approach is that it does not work properly in non-transfer learning settings due to the lack of abundant data for training the surrogate throughout the HPO process. It is possible however to design another experimental protocol where a transition model is trained to estimate a partially observable search space on a particular dataset ( i.e. no transfer learning required), however, planning and navigation would have to be done in the unexplored search space.

