# OpenReview forum: "Improving Hyperparameter Optimization by Planning Ahead"
_ICLR.cc/2022/Conference — ICLR 2022 Submitted_

### Official Review · Reviewer_8eE4 · 2021-11-01

**Correctness:** 3
**Technical Novelty And Significance:** 2
**Empirical Novelty And Significance:** 2
**Recommendation:** 3
**Confidence:** 4

**Main Review:**

Strengths:
+ The paper proposes a novel search strategy based on model predictive control for HPO by utilizing the existing meta-data.
+ Empirical study highlights promising results compared to several competitors.

Weaknesses:
- The proposed approach does not conceptually match a MbRL problem and the MDP formulation seems unnecessary.
- Some technical contents in the paper are not well supported.
- The paper lacks a clear focus and the clarity of the overall approach should be improved.
- There is not enough details on the approach and experiments to reproduce the results.


Comments:
- One of the main concerns about the paper is the MDP formulation and characterizing the approach as planning in a MbRL framework, when there is no structured way to learn a policy and the setup does not conceptually match an RL problem. It seems that the approach will work just fine without mentioning the MDP and MbRL as it only predicts the transition function, in which, the next state is a target value (e.g., validation loss for a given HP configuration) that can be learned differently (vie supervised learning), that is a bit intangible in the RL context.

- The planning strategy selects k samples without exploiting the orders between different HPs, evaluates their immediate improvement (using the trained transition function), and picks the best-performing one, which is more of a search scheme rather than planning ahead, and resembles Monte-Carlo tree search or a contextual bandit approach.

- The transition function is trained to directly learn the HPs and is not used in the planning as a dynamic model.

- The literature on HPO is not properly covered, there are several works, particularly in the bandits domain, that are not discussed:
*Li et al., A novel bandit-based approach to hyperparameter optimization, 2018
*Falkner et al. BOHB: Robust and efficient hyperparameter optimization at scale, 2018
*Tavakol et al, HyperUCB: Hyperparameter Optimization using Contextual Bandits, 2019

- The authors describe the HPO as a sequential decision-making problem, while based on my knowledge, this is not very common, and they also do not provide a supporting evidence or citation for this claim. Additionally, their approach, despite mentioned otherwise, does not exploit orders in selecting the next HPs.

- It is not completely clear from the paper how aggregating the outputs of the ensemble model relates to an “excellent” estimation of uncertainty mentioned in section 5.1.

- The clarity of the paper needs improvement as the ideas are not well organized and several parts mentioned in the paper, e.g., algorithm 1 and some details about the data and experiments could be better explained. They are also not not properly referenced.

- The link to transfer learning is unclear to me.

- For a paper that is weighted more toward an empirical paper, the experiments are small-scale and the approach is not outperforming all the baselines as stated at the beginning of the paper.

- The state space is of a higher-order Markov and is growing with time t. Could you elaborate how you deal with generalization in such a large state space and why it needs to also include the loss (in addition to HP configurations)?

- The authors do not provide a detailed overview of the optimization process, e.g. in Figure 2, what is the objective function, how the loss is optimized, etc.

- It is not completely clear how the dynamic is evolved. For instance, Eq. 6 shows that the state contains the previous HPs and their respective loss, and the model predicts the next state (\hat(\lambda)). How this works for t+1? Do you compute the true validation loss for the current HP and replace it in the next state or you keep using the predictions only?

- The paper could benefit from a proofread.


Questions:
- Could you clarify why methods such as BHBO are not considered as baselines?
- Could you explain the “rank” metric in more detail?
- What is the difference of “MPC-X” and “MPC-X (vanilla)” in figure 4?

**Summary Of The Paper:**

The paper presents a novel transfer learning approach to hyperparameter optimization (HPO) by formulating the problem in a model-based reinforcement learning (MbRL) framework. In this setting, the transition function serves as a surrogate model for learning the validation loss of any black-box ML model which is trained via an ensemble of probabilistic neural networks. The authors further introduce a lookahead search strategy to sample trajectories for a model predictive control problem. Experiments demonstrate that the proposed method outperforms several competitors on a few meta-datasets for HPO tasks.

**Summary Of The Review:**

The paper contributes to the filed of automated ML by proposing a novel approach to find optimal HP configurations. However, the ideas contains several issues in terms of clarity, significance, and correctness of the claims, and I thus vote for a reject.

---

> ### Author Response · Authors · 2021-11-13
> **Rebuttal [Part 1/2]**
>
> We thank the reviewer for his feedback and for expressing his concerns. Please find below our rebuttal to your **Comments**.
>
> - We kindly disagree on the fact that there is no structured way to learn a policy. In HPO, the objective is to maximize the generalization performance of some model (reward) by finding the optimal hyperparameter configuration (action). This is typically achieved by modeling the response surface (environment) with observed samples and using an acquisition function (policy) to suggest better samples for evaluations. This is a sequential decision-making process that can be represented as an MDP as described in Section 4. In our formulation, we aim to learn a general model of the environment, via supervised learning as is the case in any MbRL approach, which consequently allows for better planning and navigation of the response surface on **unseen tasks** to improve HPO.
>
> - The mentioned policy is K-Random Shooting which is well-established in the reinforcement learning community and is competitive against more complex policies  [1]. We believe that this does not undermine the novelty of the approach but rather shows how a simple policy can be effective and motivates research in improving HPO via MbRL and planning.
>
> - There seems to be a misunderstanding here. The transition function is indeed used for planning. As pointed out in Equation 8, the transition function estimates the only missing component from the next state, which is the response. When sampling trajectories for K-random shooting, the rollout of responses, and consequently the estimated reward is obtained from the learned transition function.
>
> - HyperBand [2] and its variations, including BOHB, are solutions for multi-fidelity HPO, whereas we design our solution for general black-box optimization. The entails that our methods, including the baselines, have no access to intermediate evaluations which are required in the case of bandit-based strategies, and thus are not suitable for comparison.
>
> - We are not the first to claim HPO as a sequential decision-making problem. HPO is conventionally posed as a Sequential Global Optimization problem [3], where a surrogate is iteratively fit to the history of observed samples, and an acquisition function suggests new candidates accordingly. The process is repeated until a pre-defined budget is exhausted.
>
> - We use ensembles of probabilistic neural networks to model uncertainty because it has proven to outperform other approaches, e.g. Bayesian modeling, on regression tasks, and in estimating more complex tasks in reinforcement learning [4].
>
> - Clarity: We will happily clarify any unclear aspects and add appropriate references if pointed out.
>
> - Link to transfer learning: The transition model is trained to estimate the response of a collection of source tasks jointly through meta-learning, Algorithm 1. After the model has been trained, we perform HPO on an unseen target task, where no (lambda, response) pairs of that task have been observed during the training process. We distinguish between the vanilla approach where during the rollout on the target task, no fine-tuning is done upon observing new hyperparameters and their responses. For the non-vanilla approach, we fine-tune our transition model based on the new observations on the target task, in the same fashion that all baselines follow. The proposed ablation serves to highlight the strong generalization performance of the transition model (vanilla approach) and to investigate the impact of fine-tuning. We noticed that the distinction is missing in the paper and will update the draft accordingly.
>
>
> References:
>
> - [1] Wang, Tingwu, et al. "Benchmarking model-based reinforcement learning." arXiv preprint arXiv:1907.02057 (2019).
> - [2] Li, Lisha, et al. "Hyperband: A novel bandit-based approach to hyperparameter optimization." The Journal of Machine Learning Research 18.1 (2017)
> - [3] Bergstra, James, et al. "Algorithms for hyper-parameter optimization." Advances in neural information processing systems 24 (2011).
> - [4] Lakshminarayanan, Balaji, et al. "Simple and scalable predictive uncertainty estimation using deep ensembles." Advances in neural information processing systems 30 (2017).

---

> > ### Author Response · Authors · 2021-11-13
> > **Rebuttal [Part 2/2]**
> >
> > - In our experiments, LookAhead MPC-X outperforms the baselines in terms of rank. The average rank represents the mean across tasks of the ranks of competing methods computed using the test accuracies of the best configuration until the t-th trial and is a common metric in HPO [5, 6, 7]. LookAhead MPC-X also leads to better regret in most cases. We report LookAhead MPC-3 and LookAhead MPC-5 for comparison, however, we invite the reviewer to notice that either one of these methods is either first or second compared to the baselines.
> > - We use the validation set of each meta-dataset to identify the best stopping position through the optimization process to ensure good generalization performance. Formally, after each epoch, we randomly sample state-action pairs and evaluate their performance using negative log-likelihood, Equation 8. We append the loss to the configuration lambda to better capture the interaction between these covariates.
> >
> > - We did not want to over-complicate figure 2. The optimization process is presented in the Appendix, Algorithm 1, and the objective function is the negative log-likelihood, Equation 8.
> >
> > - Following up on your example, during planning, the state at t+1, i.e. (\hat(\lambda)_{t+1}) would be estimated based on the estimated state at t. Once an action is selected, the \hat(\lambda) is replaced with the actual validation performance.
> >
> > - We would like to address any additional concerns that you might have with regards to the structure of the paper or grammar. Any additional suggestions with regards to the proposed method are more than welcome.
> >
> > References:
> >
> > - [5] Arango, Sebastian Pineda, et al. "HPO-B: A Large-Scale Reproducible Benchmark for Black-Box HPO based on OpenML." NeurIPS Benchmark and Dataset Track (2021)
> > - [6] Wistuba, Martin et al. "Two-stage transfer surrogate model for automatic hyperparameter optimization." Joint European conference on machine learning and knowledge discovery in databases. (2016)
> > - [7] Feurer, Matthias et al. "Scalable meta-learning for Bayesian optimization using ranking-weighted gaussian process ensembles." AutoML Workshop at ICML. Vol. 7. 2018.

---

### Official Review · Reviewer_cKwe · 2021-11-01

**Correctness:** 2
**Technical Novelty And Significance:** 2
**Empirical Novelty And Significance:** 2
**Recommendation:** 3
**Confidence:** 4

**Main Review:**

Detailed comments:
1. The main claimed contributions of the paper are that it formulates HPO as an MDP, and that solving it using MPC with some lookahead is better than greedy selection. The authors themselves mention in Sec 4 that Jomaa et al. (2019) and Volpp et al. (2020) have dealt with the same setting before. The fact that non-myopic selection strategies perform better is not novel - this is effectively an active search problem, and it has been shown that non-myopic strategies perform better greedy selection (e.g. see Garnett et al (2012)). Based on this, I don't find the contributions of the paper to be particularly novel.

2.  I found the method quite hard to follow. The initial part of the paper reads gives the impression that a full transition function that maps SxA -> S is being learnt. Only later do we get that only the response is being learnt here. This is what existing surrogate functions do as well. As the authors mention, Jomaa et al. (2019) also uses the same formulation, with the one difference being they use an LSTM to model it whereas the author of this paper follow the deep sets formulation. With the mention of MPC throughout, I was under the impression that the method employs some kind of non-random policy to guide the search (as I would expect from standard MPC). Only later does it become clear that the lookahead MPCis effectively a tree search where the tree is being generated picking hyperparameters (the actions in this formulation) at uniformly randomly. The authors include the the dataset in the state space. How is this being represented?

3. I am not very clear on experimental details. Particularly, what exactly is the training datasets being used - I understand that there are 80 datasets in each training split, but how many (lambda, loss) pairs are there for each dataset? How many (lambda, loss) pairs are budgeted for each dataset in the test set? In Sec 6.5 it is stated `We report ... over 5 runs for 50 trials with three different seeds per run`. What is a run and a trial here?

4. The authors compare the results of lookahead at 3 vs 5 steps and it seems like there is no clear winner. I would like to see the results for 1 step lookahead, i.e. greedy selection, and if the lookahead actually helps here at all.

5. I find some of the assertions in the paper to be quite baffling - I am not sure if this is just due to poor wording or some fundamental misconception about MBRL. For example: `In MbRL the objective is to train a transition model to approximate an underlying transition function via interactions with an environment governed by some policy`.  The objective of MBRL is the same as model free RL - its just that that it achieves it differently. By augmenting the real experiences with simulated experiences, MBRL hopes to improve on sample efficiency over model free RL. `HPO … can be seen as a special use-case of model-based reinforcement learning developed under the guise of some idiosyncratic terms.` What is the basis of this statement?

Garnett et al (2012) - Garnett, Roman, Krishnamurthy, Yamuna, Xiong, Xuehan, Schneider, Jeff, and Mann, Richard. Bayesian optimal active search and surveying. ICML, 2012.

**Summary Of The Paper:**

The paper deals with the problem of hyperparameter selection. It formulates the problem as an MDP, and solves it using model predictive control with a short lookahead.

**Summary Of The Review:**

I found the paper quite difficult to follow in some places, with parts of the method and experimental set ups not adequately discussed. I feel that adding an algorithm of the proposed method and signposting it to each section would go a long way towards clarifying the method. I would be willing to increase my rating if these issues get addressed, but even then I think the paper lacks sufficient novelty to elicit a strong recommendation from me.

---

> ### Author Response · Authors · 2021-11-13
> **Rebuttal [Part 1/2]**
>
> We would like to thank you for your detailed review and offer the following answers:
>
> 1.  We kindly have a different view on the novelty aspect. With regards to the formulation of HPO as an MDP, you are right, as pointed out in the paper, there have been previous attempts in framing HPO within a reinforcement learning context. However, we are the first to point out that HPO can be considered a particular case of model-based reinforcement learning that can be solved with a simple policy. Both mentioned attempts try to learn a transferable policy, whereas we focus on learning the transition model and showcase the effectiveness of a simple planning policy in comparison. To the best of our knowledge, we are also the first to propose a unique non-myopic selection strategy for HPO. We acknowledge the proven performance of non-myopic strategies in active search, however, similar to HPO, active search represents yet another problem setting where Bayesian Optimization can be leveraged towards a meaningful solution.
>
> 2. You are right, here we only learn the response (similar to other surrogate functions). However, this is not uncommon in the context of model-based reinforcement learning. We refer the reviewer to [1,2,3] as a small sample of algorithms that do not learn to map the state directly but learn to indirectly approximate the state, similar to what we do. We would like to note that we mention this workaround early in the paper to prevent any confusion. Jomaa et al (2019) also learn to predict the response, however, they propose a Deep-Q learning, i.e. model-free solution, that is not competitive compared to other baselines as reported by the paper. Following the proposed MPC approach presented in [4], we state in Section 5 that we use random shooting as our policy, and the definition of MPC is upheld. MPC is a feedback scheme in which an optimal control problem is solved at each time step [5]. In this paper, we aim for a simple policy without additional computation or gradient requirements to highlight the power of planning and motivate future research in that direction. The dataset is part of the state, Equation 3, as an indicator to which response function $\ell^{(D)}$ should be queried for a particular hyperparameter, Equation 4. Another direction, currently not explored in this paper is to represent the dataset based on its meta-features via Dataset2Vec [6], and effectively condition the surrogate on these meta-features.
>
> 3. The details of the meta-dataset can be found in the Appendix. We use three different **meta-datasets** with 120 datasets in total. In a 5-fold split, we use 80 of these datasets for training, 16 for validation, and 24 for testing. Layout Md-v2, Regularization and Optimization Md-v2 contain 324, 288, and 432 unique configurations, i.e.  (lambda, loss) pairs. All configurations are budgeted for the test set during inference. However, as we also point out in Equations 11 and 12, during trajectory sampling, we only use the estimated response and not the true response for planning. The true response is appended to the state, i.e. observed, only when the hyperparameter, i.e. action, is selected by the policy. Following the same experimental protocol of HPO, each run represents a hyperparameter optimization experiment based on a different random seed, where all the baselines are initialized with the same seeds. The experiment runs for 50 trials, which in reinforcement learning terms is a rollout of 50 actions. The whole process is run 5 times (with different seeds every time).
>
> 4. We assume that you are referring to MPC-1, which does not involve lookahead (essentially pure greedy selection). Lookahead can only be employed with random shooting when the horizon is greater than 1. We refer the reviewer to the ablation, Figure 4, where MPC-1 is either on par or worse than LookAhead MPC-X. There it can be seen that LookAhead does indeed help compared to  MPC without lookahead, i.e. selecting the first action and compared to greedy MPC-1.
>
> References:
>
> - [1] Nagabandi, Anusha, et al. "Neural network dynamics for model-based deep reinforcement learning with model-free fine-tuning." 2018 IEEE International Conference on Robotics and Automation (ICRA). IEEE, 2018.
> - [2] Thanard Kurutach, et al. “Model-Ensemble Trust-Region Policy Optimization” 2018 International Conference on Learning Representations (ICLR)
> - [3] Clavera, Ignasi, et al. "Model-based reinforcement learning via meta-policy optimization." Conference on Robot Learning. PMLR, 2018.
> - [4] Kurtland Chua et al. “Deep reinforcement learning in a handful of trials using probabilistic dynamics models”. Neural Information Processing Systems (NeurIPS) 2018
> - [5] Richards, Arthur George. Robust constrained model predictive control. Diss. Massachusetts Institute of Technology, 2005.
> - [6] Jomaa, Hadi S., Lars Schmidt-Thieme, and Josif Grabocka. "Dataset2vec: Learning dataset meta-features." Data Mining and Knowledge Discovery (2021)

---

> > ### Author Response · Authors · 2021-11-13
> > **Rebuttal [Part 2/2]**
> >
> > 5. You are right, and we will correct this ambiguity and refine the explanation in the updated draft. In MbRL, the model of the environment is trained to improve sample efficiency, whereas the overall objective is to maximize the reward. As a use-case of MbRL, the objective of HPO is to maximize the generalization performance of some model (reward) by finding the optimal hyperparameter configuration (action). This is typically achieved by modeling the response surface (environment) with observed samples and using an acquisition function (policy) to suggest better samples for evaluations. We hope that this better illustrates the similarities.

---

### Official Review · Reviewer_yo76 · 2021-11-01

**Correctness:** 3
**Technical Novelty And Significance:** 3
**Empirical Novelty And Significance:** 2
**Recommendation:** 5
**Confidence:** 2

**Main Review:**

## Strengths

- Clearly described connection between HO and MDP, which was interesting (though perhaps not entirely original, given Jomaa et al, 2019).
- To my knowledge, this is the first work considering planning in HO, and results demonstrate it can be beneficial.
- Good experimental results in comparison to baselines, and a thorough set of baselines appear to be considered.


## Weaknesses

- Related/prior work could be discussed a bit more clearly in the context of this paper. To me it was unclear how Jomaa et al (2019)'s approach differs to this paper? How does the MDP in that work compare to that presented here? The description in Section 4 was not totally clear to me. In addition, directly mentioning how this work differs to others in Section 2 Related Work would help a lot to contextualize the study.
- Would help clarity significantly to have the Algorithm description (or at least a summary) in the main text, rather than appendix. I found it hard to understand the training procedure without this.
- Some pseudocode highlighting how the state representation evolves during MPC/planning/trials would be helpful, in the appendix. This would make it very clear how this approach differs from some existing ideas that do not use planning.
- I understand the motivation for the use of ensembles, but it would help a lot to have an ablation over this to understand its importance empirically also.  Since this is one of the stated contributions, such an ablation would help support the claims made in the introduction. Are there other strategies that could have been tried here? Perhaps a single probabilistic NN would perform well too?
- Could you clarify why some baselines were not considered, such as Jomaa et al (2021b)?
- Minor: Perhaps sections 5.3 and 5.2 could be swapped in order to make the narrative a bit clearer, and conclude the discussion on dynamics model training before addressing planning.
- Minor: Tables 1 and 2 could be made clearer by highlighting that 15,33,50 refer to num. of trials

**Summary Of The Paper:**

This paper proposes a method for hyperparameter optimization (HO) drawing inspiration from model-based RL (MBRL). They recast the HO problem in a markov decision process (MDP) formulation. This formulation permits the use of MBRL methods to solve the HO problem.

Concretely, having reframed HO using this MDP formulation, they propose to: (1) train an ensemble of probabilistic neural networks to model the transition function (using first order meta learning), which maps a set consisting of previous hyperparam-response value pairs and the current hyperparam setting to the associated response value (eg val loss); (2) use this transition model in Model Predictive Control (MPC) with random shootouts to obtain an effective hyperparameter setting. Given the nature of their MDP and the fact that successive states are independent of one another, they use a variant of MPC that selects the best hyperparameter setting (or action) seen (rather than use the first action, as is common in MPC).

In experiments, they demonstrate their method improves on various baselines on three different settings. They perform some ablation studies showing the importance of finetuning their model on the target domains and the importance of planning.

**Summary Of The Review:**

Overall this paper is interesting and has I think some original ideas. I have several questions about related works and importance of aspects of the method (e.g. the ensembles) that would be good to get answered. If the authors are able to do this, I would consider raising my score.

---

> ### Author Response · Authors · 2021-11-17
> **Rebuttal**
>
> Thank you for your recommendation and we hope the following addresses your concerns.
>
> - In Section 2 we briefly mention the related work in HPO, and in Section 4 we elaborate more on the prior work that uses RL for HPO. We have updated the Section 4 for better readability.
>
> - Due to space limitations, the training algorithm is made available in the Appendix. We have added 2 **additional algorithms** to the Appendix, which include a simulated rollout that highlights how the state evolves given a transition model, and the overall MetaPETS algorithm.
>
> - Based on your suggestion, we have now added an additional Ablation to the Appendix D, with a reference from Section 6.6. We investigate the impact of using varying dynamics, including re-sampling a **single** probabilistic neural network on planning.
>
> - We focus on the baselines that **do not require any additional information** aside from the hyperparameters and their responses for a fair comparison. Jomaa et al 2021b use the primary dataset distribution as part of the model.
>
> - We have  swapped the mentioned sections as per your comment.
>
> - We have made Tables 1&2 clearer by adding **Trials** to identify what the numbers 15,33, and 50 represent.
>
> Please feel free to let us know if any questions or concerns still persist, that would prevent you from raising your score toward acceptance.

---

### Official Review · Reviewer_Vs7P · 2021-11-02

**Correctness:** 3
**Technical Novelty And Significance:** 3
**Empirical Novelty And Significance:** 4
**Recommendation:** 8
**Confidence:** 3

**Main Review:**

The authors provide a very good overview of HPO work and position their work well.

In the related work section, they miss reporting other related work coming from RL, which is in turn briefly discussed in section 4. This is not very concerning, since the authors later compare to this body of work in the experiments section. However, it would be fair to acknowledge this precedent of establishing an equivalence between HPO and MRL more clearly in the related work section in order to correctly establish the novelty of the presented approach.

The authors propose and use an ensemble of probabilistic neural networks as surrogates. The authors make a case for their choice of using bootstrapped ensembles of probabilistic neural networks. However, they don't evaluate their efficiency or stability (in the context of non-transfer scenarios). In the transfer learning setting, the use of this type of surrogate is clear. However, in a non-transfer learning setting, the training data is either small or demands computationally expensive evaluation of a large number of candidates. At some point, this brings into question the utility of a surrogate at all, since it would make sense to actually train and evaluate complete models.  Shedding a light in this direction and possible limitations would be beneficial.



**Summary Of The Paper:**

The authors establish the equivalency of hyperparameter optimization (HPO) and model-based reinforcement learning (MRL). On one hand, hyperparameter optimization is seen as an optimization of a sequence of actions (hyperparameter candidates) that improves a reward function. On the other hand, planning replaces the acquisition function of HPO. The sequence is constructed by a transition function driving the optimization towards its extremum. The transition function can be represented by a Markov Decision Process, while the reward function can be approximated by a surrogate. Within this frame, they explore the effect of planning on the performance of this approach on the task of hyperparameter optimization.

**Summary Of The Review:**

The paper is well written, the work is presented in-depth and clearly. I believe that the presented work is solid and will contribute as a point for constructive discussion between the HPO and RL communities.

---

> ### Author Response · Authors · 2021-11-17
> **Rebuttal**
>
> We would like to thank the reviewer for his interest in our paper and for his recommendation.
>
> - As you have rightfully pointed out, we describe the directly related prior work in Section 4 for completeness.
>
> - We have added a Limitations Section in the Appendix to address a different non-transfer learning scenario.
>
> We are happy to address any further remarks.

---

> > ### Comment · Reviewer_Vs7P · 2021-11-18
> > **Related work**
> >
> > Considering also the comments made by reviewers yo76 and cKwe, it would be better to extend section 2 instead and make clear the distinction between your work and the related work.

---

### Author Response · Authors · 2021-11-17
**Summary of Changes**

We thank all the reviewers for their constructive criticism of this paper. Aside from the individual rebuttals, we present here a summary of the changes in the updated draft:

- Fixed the ambiguity surrounding Model-based reinforcement learning and clarified its relationship to HPO.
- Adjusted the notation of the MDP in Section 4 and throughout the paper for better readability.
- Added 2 new algorithms to the Appendix that describe how a simulated rollout can be generated as well as the overall MetaPETS Algorithm.
- Clarified some terms, e.g. MPC (vanilla)
- Introduced an addition Ablation in Appendix D where we compare the ensemble approach with single probabilistic neural networks and bootstrapping a subset of the models with every trial.
- Identified the limitation of the approach in non-transfer learning settings, Appendix E.
- Uploaded the source code to the supplementary material. The saved checkpoints will be made available upon publication.

Please feel free to let us know if you have any more questions or concerns.

---

### Decision · Program_Chairs · 2022-01-20

**Decision:**

Reject

**Comment:**

This paper proposes an algorithm for hyperparameters optimization that exploits a formulation as an MDP and thus makes use of a model-based reinforcement learning approach.

The formulation of HPO as an MDP although not novel (Jomaa et al. is not the only one to have considered this case, and the connection between the two was already known in the community) is indeed an interesting topic that could be impactful for the community. Unfortunately, the current manuscript is not providing much new insight into the topic.

After carefully reading the paper, I agree with Reviewers cKwe and 8eE4 that the current manuscript has several points of concern:
1) the formulation as a sequential decision-making problem is not fully elaborated
2) lacking comparison to look-ahead (i.e., non-myopic) HPO algorithms (there is plenty of literature on Bayesian Optimization for doing this). This also makes it difficult to understand if the performance benefits come from the look-ahead or from the MDP formulation
3) the writing is generally understandable, but some of the important design choices and details of the algorithms are not easy to find in the manuscript -- improving the clarity of the text would be very beneficial.

I encourage the authors to incorporate the feedback from the reviewer and to polish this paper into the shiny gem that it deserves to be.

Suggestions:
- The MDP formulation for HPO might actually prove very beneficial for hyperparameters control (i.e., dynamically adjusting parameters during the learning process) where there is a real transition function rather than hyperparameters optimization. Might be worth reading https://arxiv.org/abs/2102.13651 which attempts to do hyperparameters control in the context of MBRL.
- Adding better visuals to explain formulation and algorithm might go a long way.
- Tables 1 and 2 could be replaced by learning curves for a more intuitive way of visualizing the results.